# Combining Protein Content and Grain Yield by Genetic Dissection in Bread Wheat under Low-Input Management

**DOI:** 10.3390/foods10051058

**Published:** 2021-05-11

**Authors:** Junjie Ma, Yonggui Xiao, Lingling Hou, Yong He

**Affiliations:** 1Institute of Environment and Sustainable Development in Agriculture, Chinese Academy of Agricultural Sciences, Beijing 100081, China; caasmajunjie@163.com; 2Institute of Crop Sciences, Chinese Academy of Agricultural Sciences, Beijing 100081, China; xiaoyonggui@caas.cn; 3School of Advanced Agricultural Sciences, Peking University, Beijing 100871, China

**Keywords:** bread wheat, protein content, grain yield, genetic dissection, low-input management

## Abstract

The simultaneous improvement of protein content (PC) and grain yield (GY) in bread wheat (*Triticum aestivum* L.) under low-input management enables the development of resource-use efficient varieties that combine high grain yield potential with desirable end-use quality. However, the complex mechanisms of genotype, management, and growing season, and the negative correlation between PC and GY complicate the simultaneous improvement of PC and GY under low-input management. To identify favorable genotypes for PC and GY under low-input management, this study used 209 wheat varieties, including strong gluten, medium-strong gluten, medium gluten, weak gluten, winter, semi-winter, weak-spring, and spring types, which has been promoted from the 1980s to the 2010s. Allelic genotyping, performed using kompetitive allele-specific polymerase chain reaction (KASP) technology, found 69 types of GY-PC allelic combinations in the tested materials. Field trials were conducted with two growing season treatments (2018–2019 and 2019–2020) and two management treatments (conventional management and low-input management). Multi-environment analysis of variance showed that genotype, management, and growing season had extremely substantial effects on wheat GY and PC, respectively, and the interaction of management × growing season also had extremely significant effects on wheat GY. According to the three-sigma rule of the normal distribution, the GY of wheat varieties *Liangxing 66* and *Xinmai 18* were stable among the top 15.87% of all tested materials with high GY, and their PC reached mean levels under low-input management, but also stably expressed high GY and high PC under conventional management, which represents a great development potential. These varieties can be used as cultivars of interest for breeding because *TaSus1-7A*, *TaSus1-7B*, *TaGW2-6A*, and *TaGW2-6B*, which are related to GY, and *Glu-B3*, which is related to PC, carry favorable alleles, among which Hap-1/2, the allele of *TaSus1-7A*, and Glu-B3b/d/g/i, the allele of *Glu-B3*, can be stably expressed. Our results may be used to facilitate the development of high-yielding and high-quality wheat varieties under low-input management, which is critical for sustainable food and nutrition security.

## 1. Introduction

Wheat (*Triticum aestivum* L.) accounts for 21% of the global food crop [1] and is an important food crop for human survival worldwide. By 2050, the global demand for wheat is expected to increase at a rate of 1.7% per year, while wheat grain yield (GY) is expected to increase by only 1.1% per year [2]. To reduce or avoid this deficit in global supply, new and more strategic approaches need to be explored to improve wheat GY [3]. In addition to being a major source of starch and energy, wheat is also a source of numerous vital or beneficial components for human health, particularly protein and vitamins [4]. Globally, wheat provides a greater proportion of plant protein (21%) than rice (13%) and maize (4%) combined [5]. Protein content (PC) determines the nutritional and end-use properties, mixing characteristics, and rheological properties of dough, and thus affects the efficiency of the bread-making process and product quality [6]. In response to the rising global demand for food, wheat quality has received increasing attention, and PC is an important indicator of wheat grain quality. A simultaneous improvement in both PC and GY is essential to develop a high-quality wheat industry, while ensuring food and nutrition security.

Field management is a critical way to regulate the response of wheat PC and GY to the growing season. Seasonal variations in environmental factors, such as temperature, precipitation, and sunshine hours, affect PC and GY in wheat by influencing grain starch accumulation, nitrogen uptake and translocation [7], and the metabolic processes that impact the carbon, nitrogen, and lipid contents of plants and grain [8]. The implementation of appropriate water deficit strategies and the improvement of on-farm nitrogen fertilizer management can improve water-use efficiency, promote nitrogen translocation, and increase PC and GY [9]. In general, the cultivation of high-yielding and high-quality wheat requires more water and fertilizer resources [10], and this poses several environmental conflicts. Global water scarcity is a growing problem [11], while excessive inputs of nitrogen fertilizers have exacerbated the degradation of the aquatic environment, polluted aquatic and terrestrial ecosystems, and constrained the sustainable development of society [12]. Therefore, explorations of low-input agriculture are vital to mitigate environmental pollution problems and adapt wheat production to the growing season to achieve benefits and avoid harm [13].

Different genotypes and their genetic characteristics are also important factors that affect PC and GY in wheat. In terms of GY, wheat genotypes that accumulate more dry matter through photosynthesis before flowering and translocate more of the accumulated photosynthetic products to the grain after flowering have a larger grain size and higher GY [14]. Koutroubas et al. (2012) concluded that wheat genotypes with faster pre-flowering growth rates are more likely to accumulate dry matter and nitrogen and increase GY, while wheat genotypes with excessive plant height are more susceptible to severe collapse and lower GY [15]. In terms of PC, the nitrogen uptake capacity of wheat plants from flowering to maturity was one of the main reasons for the higher PC of high PC wheat genotypes compared to medium and low PC wheat genotypes [16]. Further studies on the process of wheat grain protein accumulation found that the PC of high PC genotypes increases until maturity, while the PC of low PC genotypes undergoes a small decrease during the grain-filling process [17]. Additionally, different genotypes have been shown to respond differently to the same environmental traits [18]. Numerous studies have reported the response of different genotypes in terms of PC and GY to various meteorological conditions. For example, Tack et al. (2014) analyzed 26 years of field data from 11 locations in Kansas State and concluded that varieties from 2005 onward had higher GY than older varieties, but unfavorable meteorological conditions reduced the advantage of GY [19]. The results of Tahir and Nakata showed that, under high-temperature stress conditions, the high PC genotypes of wheat had significantly higher PC, while the medium and low PC genotypes had significantly lower PC [20]. Flagella et al. (2010) found no significant difference in GY between the two varieties studied, but the PC decreased more significantly in the low PC varieties under drought stress conditions [21].

The adaptation of wheat genotypes to different environments or growing seasons can affect the stability of their PC and GY expressions. Stability refers to the ability of genotypes to behave consistently under a wide range of environmental conditions. It is important to identify genotypes that are able to stably express important traits (e.g., GY and PC) across different environments [22]. In regards to the stability of wheat GY, PC can vary not only among genotypes, but also within the same genotype under different environmental conditions. For example, Knapp and van der Heijden [23] compared the stability of GY in wheat under low-input fertilizer management and conventional fertilizer management by using a meta-analysis based on 532 multi-year observations and concluded that the temporal stability of GY in wheat was lower under low-input management than under conventional management. These authors stressed that, in addition to focusing on environmentally friendly aspects, measures should be taken to reduce GY variability in wheat. Additionally, Würschum et al. (2016) conducted field experiments with 407 winter wheat varieties at three locations and showed that the PC trait, despite being highly heritable, is also influenced by genotype × environment interactions and is less stable, and that breeding should be based on the results of field trials at multiple locations [24]. For the relationship between GY and PC, Mosleth et al. (2020) also conducted field experiments at multiple locations in England and demonstrated that grain PC deviations, a phenomenon that some wheat genotypes consistently deviate from the negative relationship between the GY and the PC, were higher in wheat varieties bred in the UK than in varieties bred in other European countries [25]. These stability studies provide a reasonable basis for understanding the effects of genotype, management, and growing season on PC and GY in wheat. However, the limited number of wheat genotypes investigated (presumedly due to cost constraints) reduces the validity and a broad representation of the findings [26,27]. Besides, the applicability of the conclusions of previous genotype × management × growing season studies on wheat quality is severely limited by the expensive and time-consuming nature of subsequent PC measurements [28].

Recent studies have identified the importance of achieving a simultaneous enhancement of PC and GY in wheat under low-input management for the development of a high-quality wheat industry [29,30]. However, a negative correlation and trade-off relationship often exists between PC and GY in wheat, which complicates the simultaneous enhancement of both traits. For example, based on field experiments in semi-arid areas of central Anatolia, Tari (2016) found that the implementation of water deficit at the jointing stage increased PC, but significantly reduced GY in wheat [31]. Several studies have also reached similar conclusions for other cereal grains, such as maize [32], soybean [33], barley [34], and cowpea [35]. However, Brevis and Dubcovsky (2010) pointed that develop genotypes with higher N-use efficiency by increasing either the N-uptake or N-remobilization would allow modern wheat cultivars to improve PC without associated GY penalties [36]. Nevertheless, limited number of cultivars have been released that report no GY loss, which suggests that it is possible to develop wheat with both high GY and high PC using the appropriate combination of genetics [37]. For example, Kumar et al. (2011) reported that, with introgression of a major gene for high PC, progenies with significantly higher PC than their recipient parental genotypes and having no yield penalty were obtained [38]. A filed experiment conducted by Torrion et al. (2017) also showed that most genotypes that carry gene up regulating N metabolism to increase protein accumulation could maintain or increase GY as well [39]. Michel et al. (2019) also support the opinion that genomic selection can open up the opportunity to select cultivars that combine both high GY potential with high PC [40]. Based on these findings, the present study aimed to identify high-yielding and high-quality allelic combinations for the simultaneous enhancement of PC and GY in wheat under low-input management. To achieve this, we assessed the genotype × management × growing season interaction as the breakthrough point and conducted a 2-year field experiment to observe PC and GY in wheat under different management treatments using 209 wheat varieties as the study material. The identification of favorable genotypes for PC and GY under low-input management can provide information to support the simultaneous enhancement of crop GY and quality in low-input agriculture systems worldwide.

## 2. Materials and Methods

### 2.1. Field Conditions

Field experiments were conducted over 2 years (2018–2019 and 2019–2020) at the Xinxiang Comprehensive Experimental Station of the Chinese Academy of Agricultural Sciences (35°18′ N, 113°51′ E, 78 m above sea level), Henan Province. This area is characterized by a warm temperate semi-humid climate, with an annual frost-free period of ~205 days. The soil is classified as brown soil, with representative soil pH and bulk density values of 7.11 and 1.38 g/cm^3^, respectively, and a saturated water content, organic carbon content, and total nitrogen content of 36.10%, 1.70 g/kg, and 1.11 g/kg, respectively. In the study area, crops are irrigated via the pumping of groundwater. The accumulated temperature and total sunshine hours varied between the two growing seasons. The accumulated temperature in 2018–2019 growing season (2559.3 °C) was higher than that of 2019–2020 growing season (2437.1 °C). The total sunshine hours in 2018–2019 growing season was 45.6 h higher than that of 2019–2020 growing season. The total precipitation was 105.5 mm and 110.6 mm for 2018–2019 and 2019–2020 growing season, respectively.

### 2.2. Experimental Design

Populations of 209 wheat varieties were evaluated under conventional management and low-input management conditions. Each treatment was replicated twice using a randomized complete block design with a sowing density of 270 plants/m^2^. Based on the management experience of local farmers and considering the occurrence of precipitation during the grain-filling stage, the conventional management area was irrigated four times (winter water, revival water, jointing water, and grain-filling water) in 2018–2019, three times (winter water, revival water, and jointing water) in 2019–2020, while the low-input management area was irrigated once (winter water) in each of the two growing seasons. Bottom fertilizer was applied during the two growing seasons, as follows: in the conventional management area, 50 kg of compound fertilizer with 10 kg of urea was applied in winter and 15 kg of urea was applied in spring; in the low-input management area, 50 kg of compound fertilizer with 10 kg of urea was applied in winter. Pest and weed control measures were implemented in accordance with local standards for productive fields.

### 2.3. Materials

According to the grain hardness, PC, gluten strength, elongation and use, wheat can be classified into strong-gluten types, medium-strong gluten types, medium gluten types, and weak gluten types. Among the 209 varieties, strong-gluten types accounted for 29.03%, medium-strong gluten types accounted for 4.30%, medium gluten types accounted for 59.14%, and weak gluten types accounted for 7.53%. Additionally, 14.21% of the varieties were popularized in the 1980s and before, 24.04% in the 1990s, 50.82% in the 2000s, and 10.93% in the 2010s. Winter types accounted for 12.80%, semi-winter types accounted for 65.24%, weak spring types accounted for 17.68%, and spring types accounted for 4.27%. Chinese varieties accounted for 89.00%, of which the origins in China were distributed across nine provinces, and varieties from other countries accounted for 11.00%, of which the origins were distributed among six countries; the distribution of genotypes was reasonable (Figure 1). 

### 2.4. Genotyping

The DNA of wheat seedling leaves was extracted using the high-salt low-pH method [41], followed by PCR amplification. Finally, since kompetitive allele-specific polymerase chain reaction (KASP) markers for genes related to the spike number and grain per spike have not yet been developed, this study used the KASP fluorescence detector to genotype seven functional genes for grain weight (*TaCKX-D1*, *TaGASR7-A1*, *TaSus1-7A*, *TaSus1-7B*, *TaGS5-A1*, *TaGW2-6A*, and *TaGW2-6B*) and three functional genes for PC (*Glu-A1*, *Glu-D1*, and *Glu-B3*), which have a significant influence on the corresponding characters and whose KASP markers have been developed [42]. The primer sequences and amplification conditions of each gene are described in Table 1. According to the results of KASP assays, the 209 varieties contained 69 GY-PC allelic combinations (Table 2), which were rich in variety, thus ensuring genotypic diversity in this study.

The KASP assay was used to detect and distinguish alleles for the genes related to GY and PC. Allele-specific primers were designed carrying standard FAM (5′ GAAGGTGACCAAGTTCATGCT 3′) and HEX (5′ GAAGGTCGGAGTCAACGGATT 3′) tails with targeted SNP at the 3′ end. Assays were tested in 384-well formats and set up as 5 µL reaction, with 2.2 µL DNA (10–20 ng/µL) and 2.5 µL of the prepared genotyping mixture (2 × KASP master mixture and 0.056 µL primer mixture). The primer mixture included 46 µL ddH_2_O, 30 µL common primer (100 µM) and 12 µL of each tailed primer (100 µM). Allele-specific sequences and common primers (Table 1) were designed and tested, and the amplification conditions of each gene are described in Table 1. Protocols for the preparation and running of KASP reactions are given in the standard KASP guidelines. PCR amplification was performed using the following protocol: starting with 15 min at 95 °C, followed by ten touchdown cycles of 95 °C for 20 s and touchdown 65 °C–1 °C per cycle 25 s, further followed by 30 cycles of annealing (95 °C for 10 s; 57 °C for 60 s). Since the amplicon is less than 120 bp, no extension step was required.

### 2.5. Phenotyping and Statistical Analysis

Each plot harvested for the GY measurement was 4.2 m^2^ (3 m × 1.4 m). A FOSS improved NIR analyzer (Infratec TM 1241) was used to determine the PC, i.e., the crude PC of grain, by obtaining rich spectral information [43]. The NIR analyzer can obtain rich spectral information, using the near-infrared transmission technology and holographic digital grating to scan the full spectrum, according to the calibration database developed by the artificial neural network (ANN). All data were analyzed with R 3.6.2 (R Core Team, Vienna, Austria) to analyze and clarify the differences in PC and GY among different genotypes, management measures, and growing seasons. An analysis of variance (ANOVA) was performed according to a mixed linear model (lme4 package) to analyze and test for significance [44]. In the model, genotype, management, growing season and their interactions were regarded as fixed effects, and the repetitions of management measures was used as the random factor. The heritability (*H*^2^) of PC and GY in wheat was calculated using the following equation [45]:$$H^{2}=\frac{V_{g}}{V_{g}+V_{gs}/s+V_{gm}/m+V_{e}/sm}$$
where *V_g_* is the genetic variance, *V_gs_* is the interactive variance between variety and growing season, *V_gm_* is the interactive variance between variety and management measures, *V_e_* is the residual variance, *s* is the number of growing seasons, and *m* is the management level.

### 2.6. Selection of Allelic Combinations with High GY, High PC, and Stability

According to the 3-sigma rule of normal distribution, allelic combinations among the top 15.87% of all tested materials with the highest GY were identified in 2018–2019, 2019–2020, conventional management, and low-input management. From these allelic combinations, we then further excluded allelic combinations that lead to PC levels that both exceeded the mean values and met the Chinese good quality strong-gluten wheat standard (14%) in the low-input management environment in each growing season. Allelic combinations with high GY, high PC, and stable performance were identified if their growing season variance components (*SV_g_*) of PC and GY were smaller than the average of the growing season variance components of all genotypes, that is, if the stability of PC and GY were higher than the mean of all genotypes [46,47]. The *SV_g_* was calculated as follows: $$SV_{g}=\sum \frac{{\left(R_{gs}-m_{g}\right)}^{2}}{S-1}$$
where *R_gs_* is the GY or PC of genotype combination *g* in growing season *s*, *m_g_* is the mean GY or PC of genotype combination *g*, and *S* is the number of growing seasons, which is two in the present study (2018–2019 and 2019–2020).

## 3. Results

Selecting cultivars with high PC and high GY should combine management measures and growing season. Genotype, growing season, and management measures have significant effects on PC and GY in wheat. A significant negative correlation was detected between PC and GY and highly significant differences were detected among genotypes, management measures, and growing seasons. For GY, genotype had the greatest effect (45.5%), followed by the growing season (29.8%), the management condition (20.7%), and the management × growing season interaction effect (3.9%). For PC, genotype had a very great effect (87.0%), followed by the management condition (9.0%), and the growing season effect (4.0%) (Figure 2). PC was highly and negatively correlated with GY across growing seasons (S_2018–2019_ = −0.356, S_2019–2020_ = −0.381; *p* < 0.01), and this relationship was consistent under different management conditions (M_conventional_ = −0.312, M_low-input_ = −0.347; *p* < 0.01). Both PC and GY were highly heritable (*H*^2^ > 0.70) and significantly correlated across different growing seasons (r_GY_ = 0.550, r_PC_ = 0.557; *p* < 0.01). GY between different growing seasons was more strongly correlated with each other under conventional management (r = 0.539; *p* < 0.01) than under low-input management (r = 0.410; *p* < 0.01), but PC between different growing seasons was more strongly correlated with each other under low-input management (r = 0.593; *p* < 0.01) than under conventional management (r = 0.479; *p* < 0.01) (Figure 3). The GY under conventional management was higher than under low-input management, while the PC under conventional management was lower than under low-input management; the GY in the 2018–2019 growing season was higher than in the 2019–2020 growing season, while the PC in the 2018–2019 growing season was lower than in the 2019–2020 growing season (Figure 4a,b).

Allelic combinations with a high GY or PC were widely distributed. Chinese wheat varieties generally have high GY and low PC values in wheat. In this study, we defined the allelic combinations of the top 15.87% of all tested materials with high GY in every environment as the ‘high GY allelic combinations’, and defined the allelic combinations with PC values hgher than the mean of the tested materials in every environment (conventional management: 14.04%, low-input management: 14.47%, 2018–2019: 14.10%, 2019–2020: 14.39%) as the ‘high PC allelic combinations’. The high GY allelic combinations were mainly winter wheat with medium gluten and strong gluten, bred in the 1990s and the 2000s, mainly in Shandong, Jiangsu, and Henan, China (Figure 4c). The high PC allelic combinations were mainly winter wheat with strong gluten, bred in the 1980s, 1950s, and the 1940s, mainly in Australia, Italy, and Hubei, China (Figure 4d). The results indicated that the PC of strong-gluten wheat was higher than that of medium gluten wheat, that the Chinese varieties were bred with less attention to PC than other countries, and that there has been a fluctuating downward trend in the newly bred varieties in recent years.

Relatively balanced allelic combination exists that can achieve high GY, high PC, and high stability simultaneously. The allelic combinations A4B5, A1B3, A9B3, A12B13, A15B8, and A12B11 were among the top 15.87% of all tested materials with high GY, and the PC of A12B13 (conventional management: 14.34%, low-input management: 14.68%, 2018–2019: 14.51%, 2019–2020: 14.50%) exceeded the mean in all environments studied and met the Chinese good quality strong-gluten wheat standard (Figure 5). Calculations revealed that, under low-input management, the *SV_g_* of PC and GY of A12B13 (5.27, 6.55, respectively) were less than the *SV_g_* of the overall material (10.57, 27.41, respectively), which indicated that the allelic combination A12B13 is relatively stable genotype with both high GY and high PC under low-input management. Additionally, under conventional management, the *SV_g_* of PC and GY of A12B13 (2.30, 3.64, respectively) were also smaller than the *SV_g_* of the overall material (5.98, 12.00, respectively), which further indicate the potential of the allelic combination A12B13 for PC and GY development.

## 4. Discussion

Wheat GY is more susceptible to management conditions, growing season, and their interactions than wheat PC [28]. In this study, both the PC and GY of wheat were significantly affected by (i.e., dependent on) genotype, which provides a basis for the screening of superior cultivars and the identification of favorable genotypes. However, PC and GY were also affected by management conditions and growing season, although the effects of these factors were smaller than those of the genotype. The differences between management, growing season, and their interactions were much larger for GY than for PC because PC is mainly influenced by environmental factors during the grain-filling stage, whereas GY is influenced by environmental factors during the earlier growing stage [48]. For GY, the interaction effect of management × growing season was significant, which indicated that both the management conditions affected GY in the growing seasons; however, the interaction effect of genotype × management × growing season was not significant. These results indicated that the GY of the genotypes was similar under the different management conditions in the different growing seasons. The GY of wheat includes three elements: number of spikes, number of grains, and grain weight. Of these elements, grain weight is determined at grain maturity because grain formation occurs at the grain-filling stage, which is the final period of wheat development [49]. In contrast, the spike number and grain number are determined before flowering and are therefore affected by management conditions and growing season for a longer period of time than grain weight, which explains the greater influence of management conditions, growing season, and their interactions on GY [50]. For the PC, the interaction effects of management × growing season and genotype × management × growing season for PC were not significant, thus indicating that the PC of the genotypes were similar across the management conditions and growing seasons. Amino acids and peptides, the building blocks of proteins, start to translocate to the grains where they polymerize and accumulate protein after flowering [51]. These processes are, therefore, affected by the management conditions and growing seasons only after flowering, which results in the PC being less affected by management conditions and growing season compared to GY.

In crop systems, appropriate water and nitrogen input reduction could achieve high PC and GY and improve water and nitrogen use efficiency. Generally, there is negative correlation between GY and PC, a reduction in irrigation reduces GY and increases PC [52], while a reduction in nitrogen application reduces both PC and GY [53]. Nitrogen is an essential component of protein in grains; accumulated nitrogen before flowering is the primary source of grain nitrogen, and its retransport is the source of approximately 50–95% nitrogen in wheat grains [54,55]. In this study, the GY under low-input management was lower than that under conventional management, probably because the low-input conditions reduced the plant growth rate before flowering, which reduced the number of spikes and grains per unit area [56], and they also reduced the carbon supply intensity and starch accumulation in grains by limiting the green leaf area and the ability of plants to fix dry matter during the grain-filling stage [57]. The PC was higher under low-input management than under conventional management, and the reasons for this result are twofold. First, limited irrigation promotes nitrogen transport and accelerates protein accumulation [58], although limited nitrogen application reduced the pre-anthesis nitrogen accumulation under low-input management [59]. Second, the lower proportion of starch in grains is also a fundamental reason for the higher PC [52]. Recent studies have shown that low-input management can result in a higher PC at the expense of a small amount of GY. For example, results from a field trial by Ozturk and Aydin (2004) in Erzurum (Turkey) showed that compared with adequate irrigation, late water stress reduced GY by 24.0% and increased PC by 8.3% [60]. Zhao et al. (2005) conducted experiments in Xiaotangshan town, Beijing, and showed that GY could be increased by adjusting field management measures, and the PC could be increased by applying appropriate water stress at the grain-filling stage [61]. The above studies indicate that PC and GY in wheat respond differently to management measures, and proper farmland management can achieve high GY and high PC with improved water and nitrogen use efficiency. It is, however, noteworthy that the results of the above-mentioned studies differ slightly from those of the present study due to different meteorological conditions, which are discussed in the following section. 

Differences in meteorological conditions during the growing season are important causes of fluctuations in cereal crop GY and PC. In general, high temperatures result in lower GY and higher PC [62,63], and increased precipitation or irrigation results in higher GY and lower PC due to dilution [64,65]. In this study, the 2019–2020 study year was characterized by higher annual precipitation and lower annual cumulative temperatures, but more heat stress during the grain-filling stage than the conditions of the 2018–2019 study year. The GY in 2019–2020 was 16.40% lower than that in 2018–2019, probably because the high-temperature stress during the grain-filling stage reduced the leaf area, increased the maturity rate, reduced the net assimilation rate, reduced the synthesis of starch in grains, and lowered the grain weight [66]. In contrast, the PC in 2019–2020 was 1.12% higher than in 2018–2019, probably because high-temperature stress increased the content of HMW-GSs in the wheat grains, which, in combination with LMW-GS, forms the most important protein aggregate in wheat grain, glutenin macromer, through disulfide bonds [67]. Additionally, the PC may have increased due to lower starch content in the grains. Other studies have also found a close relationship between the growing season and PC and GY in wheat. Based on Stuttgart’s pot temperature control experiment in Germany, Zhang et al. (2019) suggested that the total amino acids and PC in grains increased during the growing season of post-flowering heat stress, despite the loss of crop GY [67]. Pan et al. (2006) conducted field experiments in four main winter wheat producing areas in China [68]. These authors demonstrated that the PC of high PC genotypes was related to the daily temperature difference during the growing season, that the PC of medium PC genotypes was associated with the total sunshine hours during the growing season, and that the PC of low protein genotypes was related to the annual mean temperature, total rainfall after flowering, and total sunshine hours during the growing season. The wheat types selected for the present study were mainly medium gluten genotypes belonging to medium PC genotypes, which are prone to increase the grain PC under high-temperature conditions during the grain-filling stage in 2019–2020. The present study, in combination with the above-mentioned studies, provides critical information for wheat breeding to develop crops with both high GY and high PC. 

The discovery of wheat genotypes with high PC and GY under low-input conditions facilitates the breeding of excellent cultivars. The breeding of new cultivars is an essential means to realize the synchronous increase in PC and GY in wheat under low-input management. Li et al. (2019) highlighted in a review that breeders need to consider both GY and the quality of grain to meet producer needs and market demands [69]. In the present study, the PC, as one of the important indexes of wheat quality, showed a significant and highly negative correlation with the GY for the studied materials as a whole, which was consistent with the results of previous studies [31,70]. Compared to varieties from other countries, wheat varieties from China have high GY and low PC. However, the allelic combination A12B13 showed high GY and high PC under low-input management, and its corresponding wheat varieties *Liangxing 66* (*Ji 91102* × *Ji 935031*, medium gluten) and *Xinmai 18* ((*C5*/*Xinxiang 3577*) *F3d1* × *Xinmai 9*, strong gluten) were all semi-winter varieties bred in the 2000s in the Huang-Huai winter wheat region, a crucial wheat production area with the highest sown area and GY in China [71]. 

Favorable alleles are beneficial for improving PC and GY levels in wheat. The functional genes of grain weight are closely related to GY, with the combination of favorable alleles *TaGW2-6A* and *TaGW2-6B* having additive effects on GY [72]. Among the allelic combinations with high GY in this study, the highest percentage of favorable alleles of *TaSus1-7A* and *TaSus1-7B* was 100%, followed by the favorable alleles of *TaGW2-6A* and *TaGW2-6B* (83%). The four-grain weight functional genes *TaSus1-7A*, *TaSus1-7B*, *TaGW2-6A*, and *TaGW2-6B* in *Liangxing 66* and *Xinmai 18* both carry the favorable alleles Hap-1/2, Hap-T, Hap-6A-A, and Hap-6B-1/2 [72,73], and thus, these varieties showed high GY. Correspondingly, protein functional genes were closely correlated with PC. The most favorable alleles of *Glu-A1* and *Glu-B3* accounted for 52% and 48%, respectively, of the high PC allelic combinations in this study, while the favorable genes of *Glu-D1* accounted for a relatively smaller percentage (35%). Studies have shown that the effect of *Glu-B3* on PC is more significant than that of *Glu-D1* and *Glu-A1*. Langner et al. (2017) showed that allelic variation at the Glu-B3 locus significantly affected PC in a three-year field trial of winter wheat double haploids [74]. Würschum et al. (2016), however, found no effect of *Glu-D1* and *Glu-A1* on PC in experiments conducted using 407 winter wheat varieties at three locations [24]. In the present study, *Liangxing 66* and *Xinmai 18* both carry favorable alleles (Glu-B3b/d/g/i) in the protein functional gene and thus have the potential to produce grains with high PC. 

Genotype stability is also vital to improve breeding efforts and to obtain high GY and good-quality cereals [75]. Stability is important under low-input management, where environmental variability tends to affect grains more significantly than under conventional management, and can result in a lack of stability in grain performance [47]. Stability is, therefore, extremely valuable in environments that are not conducive to wheat growth, and of great concern to producers and breeders. For example, Zaim et al. (2017) conducted field experiments with 24 durum wheat varieties in 10 environments for two years and found that cultivars screened with the highest and most stable GY were all distant crossing combinations, and that there was great potential to improve agronomic characters by using the wild gene pool [76]. Additionally, these authors demonstrated the feasibility of breeding high and stable GY wheat cultivars. Mosleth et al. [25] tested six wheat varieties at five sites in the UK over three years and found that the grains of genotype *Hereward* had a consistently high PC. Rharrabti et al. (2003) tested 10 kinds of durum wheat in three locations in Spain over two years and found that the varieties *Altar-aos*, *Jabato*, and *Waha* showed high stability in quality traits [77]. These two studies demonstrated the feasibility of breeding wheat cultivars with a stable expression of high GY and high PC. The stable expression of PC and GY in the crop may be due to the presence of alleles with stable expression. Regarding the stability of GY, Hou et al. (2014) showed that *TaSus1-7A* carrying Hap-1/2 had a higher 1000-grain weight in multiple environments, which indicated the stable expression of the grain weight trait of Hap-1/2 [73]. Regarding the stability of PC, Ruan et al. (2020) conducted a two-year trial with 162 double haploid durum wheat populations and found that the favorable allele of *Glu-B3* was stably expressed in all environments tested [78]. In the present study, wheat cultivars *Liangxing 66* and *Xinmai 18* contained both stable genes, *TaSus1-7A* and *Glu-B3,* which indicates that their PC and GY can be stably expressed under low-input management.

## 5. Conclusions

The prospect for increasing global food and nutrition security needs the cultivation of high-yielding and high-quality wheat in the context of sustainable agriculture. However, the complex mechanisms of genotype, management, and growing season, and the negative correlation between PC and GY, complicate the simultaneous improvement of PC and GY under low-input management. To identify favorable genotypes for both high PC and GY in bread wheat under low-input management, KASP genotyping and field trials were conducted on a panel of 209 wheat varieties in 2018–2020 and the pathways of the simultaneous enhancement of PC and GY in wheat under low-input management at the population and individual levels were analyzed, respectively. This study reports that genotype had a significant effect on wheat PC and GY (45.5% and 63.65%, respectively), management had a great effect on wheat PC (6.62%), growing season had a great impact on wheat GY (29.8%), and the management × growing season interaction had a highly significant effect on wheat GY (3.9%). Despite the relatively low PC of the current Chinese wheat varieties, superior genotypes with high GY and high PC exist. The grains of the allelic combination A12B13 can stably express high GY, i.e., among the top 15.87% of the tested materials with high GY (according to the 3-sigma rule of the normal distribution), and high PC, i.e., higher than the mean value of the overall material under low-input management. We furthermore report that the A12B13 cultivar carries favorable alleles for the grain weight functional genes TaSus1-7A, TaSus1-7B, TaGW2-6A, and TaGW2-6B related to GY and the protein functional gene Glu-B3 related to PC, and can stably express the alleles Hap-1/2 of TaSus1-7A and Glu-B3b/d/g/i of Glu-B3, demonstrating its potential value for wheat breeding. Overall, the results presented in this study provide an important leap in the understanding of high-yielding and high-quality in diverse wheat varieties in the framework of genotype × management × growing season. Future studies with the identification of more loci that contribute to the component of GY, e.g., kernel number per spike and spike number, and the component of PC, e.g., albumin, globulin, gliadin and glutelin, would enable more accurate selection of new genetic combinations in elite high-yielding and high-quality bread wheat varieties.

## Figures and Tables

**Figure 1 foods-10-01058-f001:**
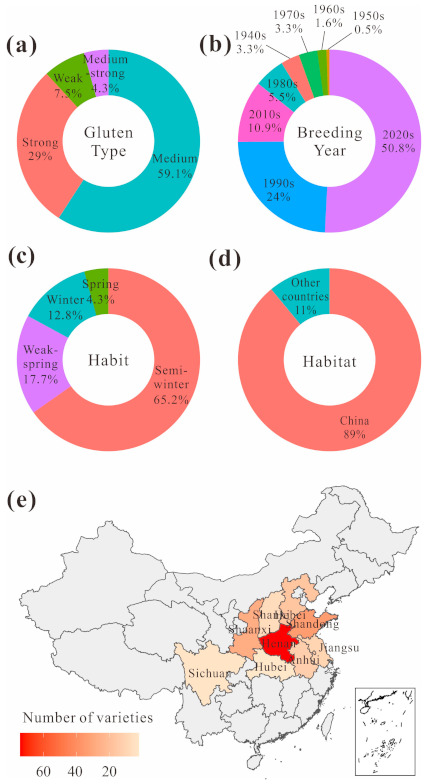
Distribution of tested varieties in terms of gluten type (**a**), breeding year (**b**), habit (**c**), and habitat (**d**), particularly among nine provinces in China (**e**).

**Figure 2 foods-10-01058-f002:**
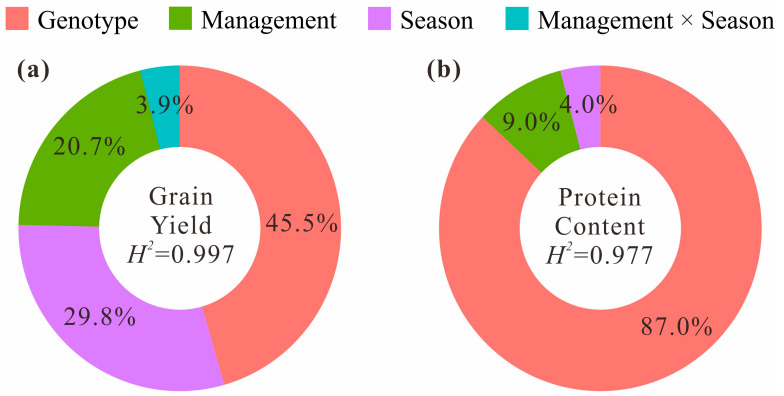
Proportion of variance components affecting the grain yield (GY) (**a**) and protein content (PC) (**b**) in wheat by genotype, management condition, growing season, and heritability (*H*^2^, generalized) in 69 allelic combinations.

**Figure 3 foods-10-01058-f003:**
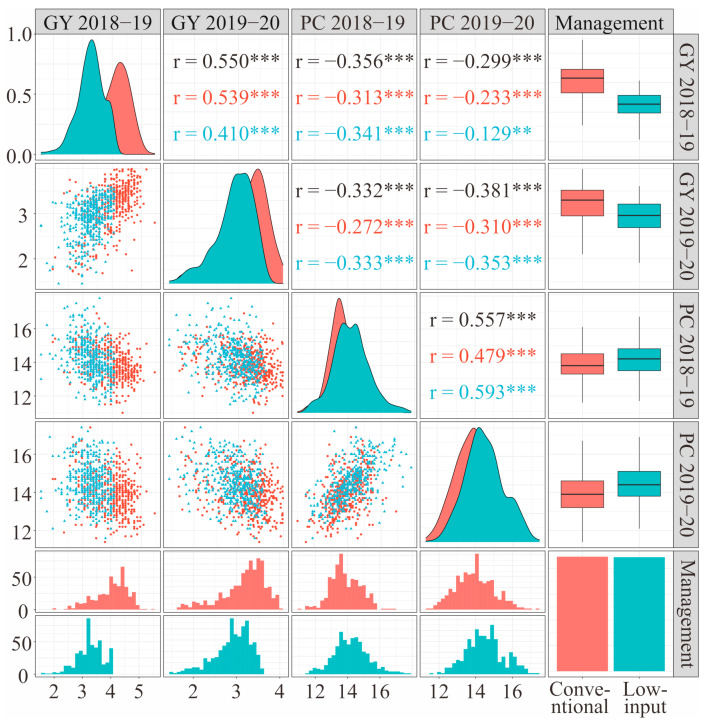
The distribution of the grain yield (GY) and protein content (PC) of different wheat genotypes of wheat in different growing seasons and under different management conditions, and the relationships between them. The *r* represents the correlation coefficient, and the color of its value is consistent with the plots in the figure. Significance level: ***, *p* < 0.001; **, *p* < 0.01.

**Figure 4 foods-10-01058-f004:**
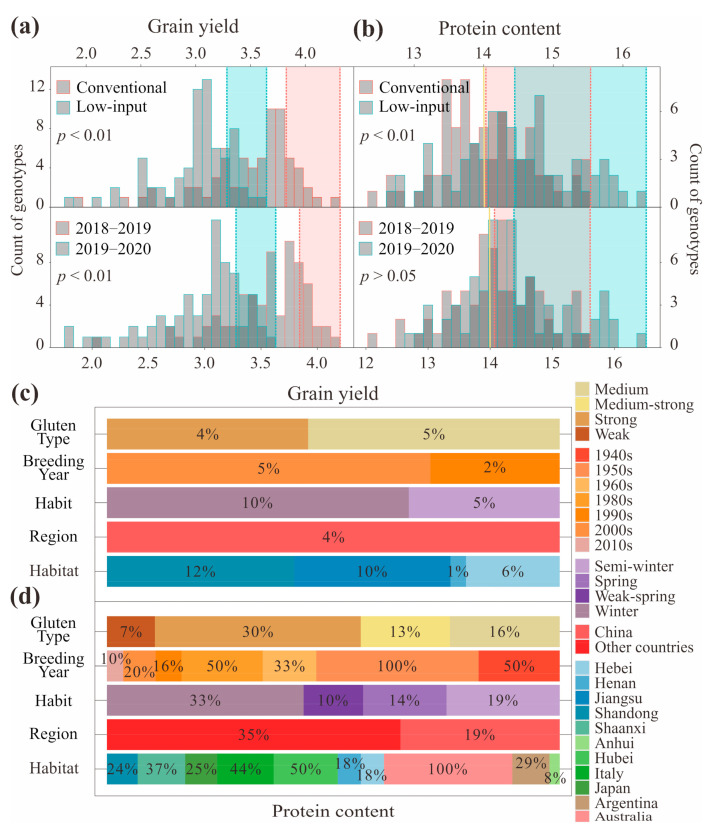
Distribution of high grain yield (GY) (**a**,**c**) or high protein content (PC) (**b**,**d**) wheat genotypes. The filled squares enclosed by virtual lines in (**a**,**b**) mark the areas where the GY falls among the top 15.87% of the tested materials with high GY or where the PC is higher than the mean of all test materials in each environment. The percentages in (**c**,**d**) represent the ratios of high GY or high PC wheat genotypes for the selection of genotypes in that area.

**Figure 5 foods-10-01058-f005:**
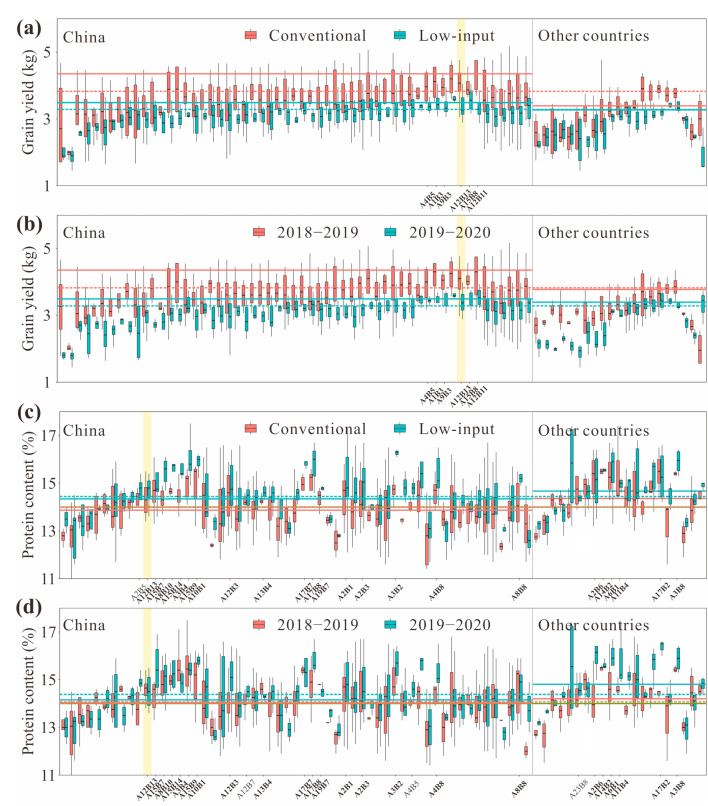
Illustration of the concepts for the simultaneous selection of grain yield (GY) (**a**,**b**) and protein content (PC) (**c**,**d**) in wheat under different management treatments and growing seasons. In (**a**,**b**), the red dotted line is the dividing line of the top 15.87% of the tested materials with the highest GY of varieties from China or other countries under conventional management or in the 2018–2019 growing season, while the blue dotted line is the dividing line of the top 15.87% of the tested materials with the highest GY of varieties from China or other countries under low-input management or in the 2019–2020 growing season; the solid red line is the dividing line of the top 15.87% of the tested materials with the highest GY under conventional management or in the 2018–2019 growing season, while the solid blue line is the dividing line of the top 15.87% of the tested materials with the highest GY under low-input management or in the 2019–2020 growing season. In (**c**,**d**), the red dotted line is the mean of PC of varieties from China or other countries under conventional management or in the 2018–2019 growing season, while the blue dotted line is the mean of PC of varieties from China or other countries under low-input management or in the 2019–2020 growing season; the solid red line is the mean of PC of all tested materials under conventional management or in the 2018–2019 growing season, while the solid blue line is the mean of PC of all tested materials under low-input management or in the 2019–2020 growing season. The solid green line is the Chinese good quality strong-gluten wheat standard (14%). For orientation, the allelic combinations that fell among the top 15.87% of the tested materials with the highest GY or PC levels higher than the mean of all test materials in each environment are labeled. The allelic combinations with either such high GY or high PC levels in every environment are bolded, but the genotypic combination that fulfil both assumptions is highlighted in the yellow vertical line.

**Table 1 foods-10-01058-t001:** The primer sequences and the amplification condition of each gene.

Trait	Gene	FAM Primer (5′-3′)	HEX Primer (5′-3′)	Common Primer (5′-3′)	Amplification Conditions
Grain yield	*TaCKX-D1*	GAAGGTGACCAAGTTCATGCTCGTCGATAGTCTCATGCATATGC	GAAGGTCGGAGTCAACGGATTATGCATGCATGCATGCGT	AACTTTTCACGGTGAACAG	Primer mixture included 46 μL ddH_2_O, 30 μL common primer (100 μM) and 12 μL of each tailed primer (100 μM). Assays were tested in 384-well format and set up as 5 μL reaction [2.2 μL DNA (10–20 ng/μL), 2.5 μL of 2XKASP master mixture and 0.056 μL primer mixture]. PCR cycling was performed using following protocol: hot start at 95 °C for 15 min, followed by ten touchdown cycles (95 °C for 20 s; touchdown 65 °C–1 °C per cycle 25 s) further followed by 30 cycles of amplification (95 °C for 10 s; 57 °C for 60 s). Extension step is unnecessary as amplicon is less than 120 bp. Plate was read in BioTek H1 system and data analysis was performed manually using Klustercaller software (version 2.22.0.5; LGC Hoddesdon, United Kingdom).
	*TaGASR7-A1*	GAAGGTGACCAAGTTCATGCTCACGGTAGAGGAGCCGGTTC	GAAGGTCGGAGTCAACGGATTATAACTGCTCACCCCCCACC	ATATGTAGGGCAGGAAGGGC	
	*TaSus1-7A*	GAAGGTGACCAAGTTCATGCTGATTTGATCCATGCCCTCTC	GAAGGTCGGAGTCAACGGATTGATTTGATCCATGCCCTCTT	CTGTCGTTCAACATCATTGTCTG	
	*TaSus1-7B*	GAAGGTGACCAAGTTCATGCTCAATTGCTTATGTTCTGTTGTATGG	GAAGGTCGGAGTCAACGGATTCAATTGCTTATGTTCTGTTGTACAT	ATGGTTATGCTTGAATGGAAGAGC	
	*TaGS5-A1*	GAAGGTGACCAAGTTCATGCTGTGCAATCTTGGACAAACATCAG	GAAGGTCGGAGTCAACGGATTGTGCAATCTTGGACAAACATCAT	AGTGCTTTGTCAACAACAGATGC	
	*TaGW2-6A*	GAAGGTGACCAAGTTCATGCTTCCCGCTCCAGCTATCTGGTGAAC	GAAGGTCGGAGTCAACGGATTTCCCGCTCCAGCTATCTGGTGAAA	TTCCCAGTCTTTGACATGTTCCGCC	
	*TaGW2-6B*	GAAGGTGACCAAGTTCATGCTTGAGATCCCGTGCAGTAGCTCG	GAAGGTCGGAGTCAACGGATTTGAGATCCCGTGCAGTAGCTCA	TGGCGTGAGCTAGGGTTTGTTG	
Protein content	*Glu-A1*	GAAGGTGACCAAGTTCATGCTAAGTGTAACTTCTCCGCAACA	GAAGGTCGGAGTCAACGGATTAAGTGTAACTTCTCCGCAACG	GGCCTGGATAGTATGAAACC	
	*Glu-D1*	GAAGGTGACCAAGTTCATGCTCGCTAATCCTGCGAGCAACAAAT	GAAGGTCGGAGTCAACGGATTGCTAATCCTGCGAGCAACAAAG	AGCCAAGGGCATGTTCTATGTCGAA	
	*Glu-B3*	GAAGGTGACCAAGTTCATGCTctgttggggttgggaaacG	GAAGGTCGGAGTCAACGGATTctgttggggttgggaaacA	agcagcagcaaccGcaaC	

**Table 2 foods-10-01058-t002:** The allelic combinations for grain yield (A, 23 combinations) and protein content (B, 15 combinations)

Haplotype	TaCKX-D1	TaGASR7-A1	TaSus1-7A	TaSus1-7B	TaGS5-A1	TaGW2-6A	TaGW2-6B	Glu-A1	Glu-D1	Glu-B3	Percent/%
A10B1	TaCKX-D1b	H1g	Hap-3/4	Hap-T	A1b	Hap-6A-A	Hap-1/2	1/2*	5 + 10	Glu-B3b/d/g/i	0.48
A11B4	TaCKX-D1b	H1g	Hap-1/2	Hap-T	A1b	Hap-6A-G/Hap-6A-A	Hap-1/2	1/2*	2 + 12	others	0.48
A12B1	TaCKX-D1b	H1g	Hap-1/2	Hap-T	A1a	Hap-6A-A	Hap-1/2	1/2*	5 + 10	Glu-B3b/d/g/i	2.42
A12B11	TaCKX-D1b	H1g	Hap-1/2	Hap-T	A1a	Hap-6A-A	Hap-1/2	null	2 + 12/5 + 10	others	0.48
A12B13	TaCKX-D1b	H1g	Hap-1/2	Hap-T	A1a	Hap-6A-A	Hap-1/2	1/2*/null	2 + 12	Glu-B3b/d/g/i	0.97
A12B15	TaCKX-D1b	H1g	Hap-1/2	Hap-T	A1a	Hap-6A-A	Hap-1/2	1/2*/null	5 + 10	Glu-B3b/d/g/i	0.48
A12B2	TaCKX-D1b	H1g	Hap-1/2	Hap-T	A1a	Hap-6A-A	Hap-1/2	1/2*	5 + 10	Glu-B3j	2.42
A12B3	TaCKX-D1b	H1g	Hap-1/2	Hap-T	A1a	Hap-6A-A	Hap-1/2	1/2*	2 + 12	Glu-B3b/d/g/i	3.38
A12B4	TaCKX-D1b	H1g	Hap-1/2	Hap-T	A1a	Hap-6A-A	Hap-1/2	1/2*	2 + 12	others	5.31
A12B6	TaCKX-D1b	H1g	Hap-1/2	Hap-T	A1a	Hap-6A-A	Hap-1/2	null	5 + 10	others	0.48
A12B7	TaCKX-D1b	H1g	Hap-1/2	Hap-T	A1a	Hap-6A-A	Hap-1/2	null	2 + 12	Glu-B3b/d/g/i	0.97
A12B8	TaCKX-D1b	H1g	Hap-1/2	Hap-T	A1a	Hap-6A-A	Hap-1/2	null	2 + 12	others	3.38
A12B9	TaCKX-D1b	H1g	Hap-1/2	Hap-T	A1a	Hap-6A-A	Hap-1/2	1/2*	2 + 12/5 + 10	others	0.97
A13B4	TaCKX-D1b	H1g	Hap-1/2	Hap-T	A1a/A1b	Hap-6A-A	Hap-1/2	1/2*	2 + 12	others	0.48
A14B2	TaCKX-D1b	H1g	Hap-1/2	Hap-C	A1b	Hap-6A-A	Hap-1/2	1/2*	5 + 10	others	0.48
A14B4	TaCKX-D1b	H1g	Hap-1/2	Hap-C	A1b	Hap-6A-A	Hap-1/2	1/2*	2 + 12	others	0.48
A15B14	TaCKX-D1b	H1g	Hap-1/2	Hap-T	A1b	Hap-6A-G	Hap-1/2	1/2*/null	2 + 12	others	0.48
A15B2	TaCKX-D1b	H1g	Hap-1/2	Hap-T	A1b	Hap-6A-G	Hap-1/2	1/2*	5 + 10	others	1.47
A15B3	TaCKX-D1b	H1g	Hap-1/2	Hap-T	A1b	Hap-6A-G	Hap-1/2	1/2*	2 + 12	Glu-B3b/d/g/i	1.47
A15B4	TaCKX-D1b	H1g	Hap-1/2	Hap-T	A1b	Hap-6A-G	Hap-1/2	1/2*	2 + 12	others	2.42
A15B7	TaCKX-D1b	H1g	Hap-1/2	Hap-T	A1b	Hap-6A-G	Hap-1/2	null	2 + 12	Glu-B3b/d/g/i	0.97
A15B8	TaCKX-D1b	H1g	Hap-1/2	Hap-T	A1b	Hap-6A-G	Hap-1/2	null	2 + 12	others	0.48
A15B9	TaCKX-D1b	H1g	Hap-1/2	Hap-T	A1b	Hap-6A-G	Hap-1/2	1/2*	2 + 12/5 + 10	others	0.97
A16B2	TaCKX-D1b	H1g	Hap-3/4	Hap-T	A1b	Hap-6A-G	Hap-1/2	1/2*	5 + 10	others	0.48
A17B2	TaCKX-D1b	H1g	Hap-1/2	Hap-C	A1b	Hap-6A-G	Hap-1/2	1/2*	5 + 10	others	0.48
A17B4	TaCKX-D1b	H1g	Hap-1/2	Hap-T	A1a	Hap-6A-G	Hap-1/2	1/2*	2 + 12	others	1.47
A17B7	TaCKX-D1b	H1g	Hap-1/2	Hap-T	A1a	Hap-6A-G	Hap-1/2	null	2 + 12	Glu-B3b/d/g/i	0.48
A17B8	TaCKX-D1b	H1g	Hap-1/2	Hap-T	A1a	Hap-6A-G	Hap-1/2	null	2 + 12	others	0.48
A18B4	TaCKX-D1b	H1c	Hap-1/2	Hap-C	A1a	Hap-6A-A	Hap-1/2	1/2*	2 + 12	others	0.48
A18B5	TaCKX-D1b	H1c	Hap-1/2	Hap-C	A1a	Hap-6A-A	Hap-1/2	null	5 + 10	Glu-B3b/d/g/i	0.48
A18B8	TaCKX-D1b	H1c	Hap-1/2	Hap-C	A1a	Hap-6A-A	Hap-1/2	null	2 + 12	others	0.48
A19B7	TaCKX-D1b	H1c	Hap-3/4	Hap-T	A1a	Hap-6A-A	Hap-1/2	null	2 + 12	Glu-B3b/d/g/i	0.48
A1B3	TaCKX-D1a	H1g	Hap-1/2	Hap-T	A1a	Hap-6A-A	Hap-1/2	1/2*	2 + 12	Glu-B3b/d/g/i	0.48
A20B14	TaCKX-D1b	H1g	Hap-1/2	Hap-T/Hap-C	A1a	Hap-6A-A	Hap-1/2	1/2*/null	2 + 12	others	0.48
A21B4	TaCKX-D1b	H1g	Hap-1/2	Hap-T/Hap-C	A1b	Hap-6A-G	Hap-1/2	1/2*	2 + 12	others	0.48
A22B9	TaCKX-D1b	H1g/H1c	Hap-1/2	Hap-C	A1b	Hap-6A-A	Hap-1/2	1/2*	2 + 12/5 + 10	others	0.48
A23B8	TaCKX-D1a	H1c	Hap-3/4	Hap-T	A1a	Hap-6A-G	Hap-1/2	null	2 + 12	others	0.48
A2B1	TaCKX-D1b	H1c	Hap-1/2	Hap-T	A1b	Hap-6A-A	Hap-1/2	1/2*	5 + 10	Glu-B3b/d/g/i	1.93
A2B2	TaCKX-D1b	H1c	Hap-1/2	Hap-T	A1b	Hap-6A-A	Hap-1/2	1/2*	5 + 10	others	0.97
A2B3	TaCKX-D1b	H1c	Hap-1/2	Hap-T	A1b	Hap-6A-A	Hap-1/2	1/2*	2 + 12	Glu-B3b/d/g/i	3.38
A2B4	TaCKX-D1b	H1c	Hap-1/2	Hap-T	A1b	Hap-6A-A	Hap-1/2	1/2*	2 + 12	Glu-B3j	0.97
A2B5	TaCKX-D1b	H1c	Hap-1/2	Hap-T	A1b	Hap-6A-A	Hap-1/2	null	5 + 10	Glu-B3b/d/g/i	0.48
A2B6	TaCKX-D1b	H1c	Hap-1/2	Hap-T	A1b	Hap-6A-A	Hap-1/2	null	5 + 10	others	0.48
A2B7	TaCKX-D1b	H1c	Hap-1/2	Hap-T	A1b	Hap-6A-A	Hap-1/2	null	2 + 12	Glu-B3b/d/g/i	3.38
A2B8	TaCKX-D1b	H1c	Hap-1/2	Hap-T	A1b	Hap-6A-A	Hap-1/2	null	2 + 12	others	1.93
A3B2	TaCKX-D1b	H1c	Hap-1/2	Hap-T	A1b	Hap-6A-G	Hap-1/2	1/2*	5 + 10	others	0.48
A3B4	TaCKX-D1b	H1c	Hap-1/2	Hap-T	A1b	Hap-6A-G	Hap-1/2	1/2*	2 + 12	others	0.48
A3B5	TaCKX-D1b	H1c	Hap-1/2	Hap-T	A1b	Hap-6A-G	Hap-1/2	null	5 + 10	Glu-B3b/d/g/i	0.48
A3B6	TaCKX-D1b	H1c	Hap-1/2	Hap-T	A1b	Hap-6A-G	Hap-1/2	null	5 + 10	others	0.48
A3B8	TaCKX-D1b	H1c	Hap-1/2	Hap-T	A1b	Hap-6A-G	Hap-1/2	null	2 + 12	others	0.48
A4B4	TaCKX-D1b	H1c	Hap-1/2	Hap-T	A1a	Hap-6A-A	Hap-1/2	1/2*	2 + 12	others	0.48
A4B5	TaCKX-D1b	H1c	Hap-1/2	Hap-T	A1a	Hap-6A-A	Hap-1/2	null	5 + 10	Glu-B3b/d/g/i	0.97
A4B6	TaCKX-D1b	H1c	Hap-1/2	Hap-T	A1a	Hap-6A-A	Hap-1/2	null	5 + 10	others	0.97
A4B7	TaCKX-D1b	H1c	Hap-1/2	Hap-T	A1a	Hap-6A-A	Hap-1/2	null	2 + 12	Glu-B3b/d/g/i	0.97
A4B8	TaCKX-D1b	H1c	Hap-1/2	Hap-T	A1a	Hap-6A-A	Hap-1/2	null	2 + 12	others	0.48
A6B1	TaCKX-D1b	H1c	Hap-1/2	Hap-T	A1a	Hap-6A-G	Hap-1/2	1/2*	5 + 10	Glu-B3b/d/g/i	0.48
A7B8	TaCKX-D1b	H1c	Hap-1/2	Hap-C	A1b	Hap-6A-A	Hap-1/2	null	2 + 12	others	0.48
A8B1	TaCKX-D1b	H1g	Hap-1/2	Hap-T	A1b	Hap-6A-A	Hap-1/2	1/2*	5 + 10	Glu-B3b/d/g/i	4.35
A8B10	TaCKX-D1b	H1g	Hap-1/2	Hap-T	A1b	Hap-6A-A	Hap-1/2	1/2*	2 + 12/5 + 10	Glu-B3b/d/g/i	0.97
A8B12	TaCKX-D1b	H1g	Hap-1/2	Hap-T	A1b	Hap-6A-A	Hap-1/2	null	2 + 12/5 + 10	Glu-B3b/d/g/i	0.97
A8B14	TaCKX-D1b	H1g	Hap-1/2	Hap-T	A1b	Hap-6A-A	Hap-1/2	1/2*/null	2 + 12	others	0.97
A8B2	TaCKX-D1b	H1g	Hap-1/2	Hap-T	A1b	Hap-6A-A	Hap-1/2	1/2*	5 + 10	others	2.9
A8B3	TaCKX-D1b	H1g	Hap-1/2	Hap-T	A1b	Hap-6A-A	Hap-1/2	1/2*	2 + 12	Glu-B3b/d/g/i	9.66
A8B4	TaCKX-D1b	H1g	Hap-1/2	Hap-T	A1b	Hap-6A-A	Hap-1/2	1/2*	2 + 12	others	13.53
A8B5	TaCKX-D1b	H1g	Hap-1/2	Hap-T	A1b	Hap-6A-A	Hap-1/2	null	5 + 10	Glu-B3b/d/g/i	0.48
A8B6	TaCKX-D1b	H1g	Hap-1/2	Hap-T	A1b	Hap-6A-A	Hap-1/2	null	5 + 10	others	0.97
A8B7	TaCKX-D1b	H1g	Hap-1/2	Hap-T	A1b	Hap-6A-A	Hap-1/2	null	2 + 12	Glu-B3b/d/g/i	2.9
A8B8	TaCKX-D1b	H1g	Hap-1/2	Hap-T	A1b	Hap-6A-A	Hap-1/2	null	2 + 12	others	1.47
A9B3	TaCKX-D1b	H1g	Hap-1/2	Hap-T	A1b	Hap-6A-A	Hap-4	1/2*	2 + 12	Glu-B3b/d/g/i	0.48

The numbers represent the respective percentages. “1” and “2*” are two subunits of Glu-A1.

## Data Availability

The data presented in this study are available on request from the corresponding author.

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
