# Peer review of "Combining Protein Content and Grain Yield by Genetic Dissection in Bread Wheat under Low-Input Management"

_foods, 2021, doi:10.3390/foods10051058_

Round 1

Reviewer 1 Report

In the manuscript titled “Combining protein content and grain yield by genetic dissection in bread 1 wheat under low-input management”, the main objective is to identify high-yielding and high-quality allelic combinations in wheat under low-input management, which could be considered quite relevant for a wheat breeding program. However, in bibliography related to this topic, there is a negative correlation between both desirable characters, for this reason, a clear justification of the possibility to identify genes related high-yielding and high-quality in these varieties should be included. Moreover, the material and method section is quite poor in the genotyping and characterization of all wheat varieties used. The plant material used in this manuscript is not properly described, considering that only strong gluten, medium-strong gluten, medium gluten, weak 30 gluten, winter, semi-winter, weak-spring, and spring types have been included. Regarding genotyping, materials and methods is not correctly included in the manuscript. No details about primer sequences and PCR conditions used. More information about NIR should be included.

Author Response

In the manuscript titled “Combining protein content and grain yield by genetic dissection in bread wheat under low-input management”, the main objective is to identify high-yielding and high-quality allelic combinations in wheat under low-input management, which could be considered quite relevant for a wheat breeding program. However, in bibliography related to this topic, there is a negative correlation between both desirable characters, for this reason, a clear justification of the possibility to identify genes related high-yielding and high-quality in these varieties should be included. Moreover, the material and method section is quite poor in the genotyping and characterization of all wheat varieties used. The plant material used in this manuscript is not properly described, considering that only strong gluten, medium-strong gluten, medium gluten, weak gluten, winter, semi-winter, weak-spring, and spring types have been included. Regarding genotyping, materials and methods is not correctly included in the manuscript. No details about primer sequences and PCR conditions used. More information about NIR should be included. 

Response: We accept these concerns and have added related supporting information for each of these sub-questions, as follows:

  • In bibliography related to this topic, there is a negative correlation between both desirable characters, for this reason, a clear justification of the possibility to identify genes related high-yielding and high-quality in these varieties should be included.

Response: We accept these concern. To justify the possibility to identify genes related high-yielding and high-quality in these varieties should be included, we add content to highlight the possibility after introducing the different accumulation mechanism of GY and PC in different genotypes (L96–L97).

  • The plant material used in this manuscript is not properly described, considering that only strong gluten, medium-strong gluten, medium gluten, weak gluten, winter, semi-winter, weak-spring, and spring types have been included.

Response: We accept these concern. The gluten types, including strong gluten, medium-strong gluten, medium gluten and weak gluten, are closely related to the protein content in the present study. For this concern, we moved the result of KASP (L211–L214) form “2.2 Experimental design and materials” to “2.3 Genotyping” (L224–L227) for clearly demonstrating the properties of materials. In addition, we also introduced other properties of materials, such as breeding year and habitat at L204–L211.

  • Regarding genotyping, materials and methods is not correctly included in the manuscript. No details about primer sequences and PCR conditions used.

Response: Agreed. We add a table (Table 1S) including primer sequences and PCR conditions.

  • More information about NIR should be included.

Response: Agreed. Description of NIR analyzer were added in the corresponding position in the text (L229–L232).

Reviewer 2 Report

Dear Authors

The study underscores the importance of balancing yield and protein content.  With yield having an inclination towards management practices, and protein content though higher, it would be better to check the glutenin and gliadin components under normal and low-input management.  Because, under low input conditions, gliadin content might have an increase and thereby the PC is higher.  This will hinder the bread making quality.  So, if possible, authors can add a table depicting the proportion of glutenin and gliadin components that gets increased to have an overall clarity in choosing the genotypes according to the environment.  

Author Response

The study underscores the importance of balancing yield and protein content. With yield having an inclination towards management practices, and protein content though higher, it would be better to check the glutenin and gliadin components under normal and low-input management. Because, under low input conditions, gliadin content might have an increase and thereby the PC is higher. This will hinder the bread making quality. So, if possible, authors can add a table depicting the proportion of glutenin and gliadin components that gets increased to have an overall clarity in choosing the genotypes according to the environment. 

Response: We agree the concern that it is feasible to determination of the effect of low-input management on the gliadin/glutenin ratio, which is worthy of attention. Actually, we will start to test gliadin/glutenin ratio based on current study soon. However, this is time and label consuming. So, our present study still focus on the crude protein content.

Reviewer 3 Report

The manuscript under review evaluated more than 200 wheat varieties that vary in both gluten types and maturity types to identify suitable genotypes for high protein and yield. Two years experiments with two management treatments showed that three varieties have high-yielding and high-quality allelic combinations. The results are useful for breeding.

Specific comments:

1 consider changing the title, the work is more like screening of cultivars for high protein and grain yield, not combining the two traits

2 It is a long Introduction, many examples were given. And there are may references (80), which occupies 11 pages. It is better to concisely summarize previous studies (rather than giving examples), point out the research needs and cite the most relevant references.

3 L167 Is there any special purpose to compare climate conditions during experimental period with long term ones? If not, I suggest to delete this part. Climate during experimental period was different from long term, if you compare, questions will be arisen about the representation of the experimental results.

4 L250 it needs more explanation here. A significant negative correlation was detected between PC and GY, then how high PC and high GY cultivar was selected?

5 L261 GY was more strongly correlated with what?

Author Response

The manuscript under review evaluated more than 200 wheat varieties that vary in both gluten types and maturity types to identify suitable genotypes for high protein and yield. Two years experiments with two management treatments showed that three varieties have high-yielding and high-quality allelic combinations. The results are useful for breeding.

Specific comments:

  1. Consider changing the title, the work is more like screening of cultivars for high protein and grain yield, not combining the two traits.

Response: We agree with this concern. After serious discussion with coauthors and comparison with literatures, we think the ‘Combining’ in our present title is more suitable for this paper, which covers meaning of both ‘identify’ and ‘simultaneously’.

Referenced title from:    

Michel, S.; Löschenberger, F.; Ametz, C.; Pachler, B.; Sparry, E.; Bürstmayr, H. Combining grain yield, protein content and protein quality by multi-trait genomic selection in bread wheat. TAG. Theoretical and applied genetics. 2019, 132, 2767-2780, doi:10.1007/s00122-019-03386-1.

  1. It is a long Introduction, many examples were given. And there are may references (80), which occupies 11 pages. It is better to concisely summarize previous studies (rather than giving examples), point out the research needs and cite the most relevant references.

Response: Agreed. We reduced some references in the Introduction section.

  1. L167 Is there any special purpose to compare climate conditions during experimental period with long term ones? If not, I suggest to delete this part. Climate during experimental period was different from long term, if you compare, questions will be arisen about the representation of the experimental results. 

Response: Agreed. We delete the meteorological data of previous years, and redescribed the climate conditions during experimental period (L168–L173).

  1. L250 it needs more explanation here. A significant negative correlation was detected between PC and GY, then how high PC and high GY cultivar was selected?  

Response: We agree with this concern, and have added more explanation (L262–L263).

  1. L261 GY was more strongly correlated with what?

Response: Agreed. We add the relevant content to the article (L275–L278), that GY between different growing seasons was more strongly correlated with each other under conventional management than under low-input management.

Round 2

Reviewer 1 Report

I thank authors to consider the comments included in my review. However, information included are not justifying correctly the main concerns that I consider in this manuscript.

Author Response

Reply to Reviewer 1 (round 2)

Once again we wish to appreciate reviewer #1 again for her/his kind remarks, constructive criticisms, and valuable time. We went through all comments carefully and made further revisions

Comments (round 2)

I thank authors to consider the comments included in my review. However, information included are not justifying correctly the main concerns that I consider in this manuscript.

  • In bibliography related to this topic, there is a negative correlation between both desirable characters, for this reason, a clear justification of the possibility to identify genes related high-yielding and high-quality in these varieties should be included (round 1).

Response: We accept this concern. We add content on the potential of developing wheat with improved protein and yield using the appropriate combination of genetics in introduction section based on new literature review. These evidence (evaluation test and meta-analysis) suggests the possibility to identify genes related high-yielding and high-quality in these varieties. The new added content is as follows:

‘However, Brevis and Dubcovsky (2010) pointed that develop genotypes with higher N-use efficiency by increasing either the N-uptake or N-remobilization would allow modern wheat cultivars to improve PC without associated GY penalties [37]. Nevertheless, limited number of cultivars have been released that report no GY loss, which suggests that it is possible to develop wheat with both high GY and high PC using the appropriate combination of genetics [38]. For example, Kumar et al. (2011) reported that, with introgression of a major gene for high PC, progenies with significantly higher PC than their recipient parental genotypes and having no yield penalty were obtained [39]. An filed experiment conducted by Torrion et al. (2017) also showed that most genotypes which carry gene up regulating N metabolism to increase protein accumulation could maintain or increase GY as well . Michel et al. (2019) also support the opinion that genomic selection can open up the opportunity to select cultivars that combine both high GY potential with high PC [41].’

  • The plant material used in this manuscript is not properly described, considering that only strong gluten, medium-strong gluten, medium gluten, weak gluten, winter, semi-winter, weak-spring, and spring types have been included.

Response: We accept this concern. We improved the structure of the article to make it easier to understand. The gluten types, including strong gluten, medium-strong gluten, medium gluten and weak gluten, are considered because they are closely related to the protein content studied in the present study. In addition, we also introduced other properties of materials, such as breeding year and habitat at L197–L203. For this concern, we have added the basis of the classification of gluten types, moved the content of ‘materials’ from ‘2.2 Experimental design and materials’ to ‘2.3 Materials’ (L204), and moved the result of KASP form ‘2.3 Materials’ (L216–L219) to ‘2.4 Genotyping’ (L229–L232) for clearly demonstrating the properties of materials.

  • Regarding genotyping, materials and methods is not correctly included in the manuscript. No details about primer sequences and PCR conditions used.

Response: We agreed this concern. We add a table (Table 1S) including primer sequences and PCR conditions. The corresponding description is as follows (L233–L247):

 ‘The KASP assay was used to detect and distinguish alleles for the genes related to GY and PC. Allele-specific primers were designed carrying standard FAM (5' GAAGGTGACCAAGTTCATGCT 3') and HEX (5' GAAGGTCGGAGTCAACGGATT 3') tails with targeted SNP at the 3' end. Assays were tested in 384-well formats and set up as 5 µl reaction, with 2.2 µl DNA (10–20 ng/µl) and 2.5 µl of the prepared genotyping mixture (2×KASP master mixture and 0.056 µl primer mixture). The primer mixture included 46 µl ddH2O, 30 µl common primer (100 µM) and 12 µl of each tailed primer (100 µM). Allele-specific sequences and common primers (Table 1S) were designed and tested, and the amplification conditions of each gene are described in Table 1S. Protocols for the preparation and running of KASP reactions are given in the standard KASP guidelines. PCR amplification was performed using the following protocol: starting with 15 min at 95°C, followed by ten touchdown cycles of 95°C for 20 s and touchdown 65°C – 1°C per cycle 25 s, further followed by 30 cycles of annealing (95°C for 10 s; 57°C for 60 s). Since the amplicon is less than 120 bp, no extension step was required.’

  • More information about NIR should be included.

Response: We agreed this concern. We add the description of NIR analyzer in principle in the corresponding position in the text (L249–L254). The description is as follows:

 ‘A FOSS improved NIR analyzer (Infratec TM 1241) was used to determine the PC, i.e., the crude PC of grain, by obtaining rich spectral information [44]. The NIR analyzer can obtain rich spectral information, using the near-infrared transmission technology and holographic digital grating to scan the full spectrum, according to the calibration database developed by artificial neural network (ANN).’
